# Flight Dispersal in Supratidal Rockpool Beetles

**DOI:** 10.3390/insects15030140

**Published:** 2024-02-20

**Authors:** Jorge Plaza-Buendía, Juana María Mirón-Gatón, Antonio José García-Meseguer, Adrián Villastrigo, Andrés Millán, Josefa Velasco

**Affiliations:** 1Ecology and Hydrology Department, University of Murcia, 30100 Murcia, Spain; jorge.plaza@um.es (J.P.-B.); juanamaria.miron@um.es (J.M.M.-G.); aj.garciameseguer@um.es (A.J.G.-M.); acmillan@um.es (A.M.); 2Division of Entomology, SNSB- Bavarian State Collection of Zoology, 81247 Munich, Germany

**Keywords:** *Ochthebius*, wing morphometry, wing loading, aspect ratio, microsatellite markers, sex-biassed dispersal, coastal temporary habitats

## Abstract

**Simple Summary:**

We studied the flight dispersal of two congeneric beetle species (*Ochthebius quadricollis* and *Ochthebius lejolisii*) living in Mediterranean coastal rockpools; temporary and fragmented habitats with extreme environmental conditions (high salinity, high temperature, and strong desiccation). We used a multi-approach (experimental flying assays, wing morphology, and genetic markers) to measure flight capacity. We found that both species had similar flight behaviour, with most individuals flying when water was heated. Females had larger body sizes and wing areas and lower wing loading than males, which suggested higher dispersal capacity. The wing shape of both species was also shown to be an efficient adaptation to flight. However, the molecular data point to passive dispersal assisted by wind at small-to-medium spatial scales (<100 km).

**Abstract:**

Flight dispersal is ecologically relevant for the survival of supratidal rockpool insects. Dispersal has important consequences for colonisation, gene flow, and evolutionary divergence. Here, we compared the flight dispersal capacity of two congeneric beetle species (*Ochthebius quadricollis* and *Ochthebius lejolisii*) that exclusively inhabit these temporary, fragmented, and extreme habitats. We estimated flight capacity and inferred dispersal in both species using different approaches: experimental flying assays, examination of wing morphology, and comparison of microsatellite markers between species. Our findings revealed that both species exhibited similar flight behaviour, with 60 to 80% of the individuals flying under water heating conditions. Notably, females of both species had larger body sizes and wing areas, along with lower wing loading, than males in *O. quadricollis*. These morphological traits are related to higher dispersal capacity and more energetically efficient flight, which could indicate a female-biassed dispersal pattern. The wing shapes of both species are characterised by relatively larger and narrower wings in relation to other species of the genus, suggesting high flight capacity at short distances. Molecular data revealed in both species low genetic divergences between neighbouring populations, non-significant differences between species, and no isolation by distance effect at the study scale (<100 km). These results point to passive dispersal assisted by wind.

## 1. Introduction

Dispersal ability, that is, the ability to move between different habitat patches, significantly influences the range and distribution of aquatic insects, especially in temporary habitats [1,2,3]. This trait has a crucial role in shaping the dynamic and genetic structure of species, impacting processes such as colonisation, gene flow, and evolutionary divergence [4,5,6]. Dispersal is also an essential process for metapopulation and metacommunity dynamics [7]. For species inhabiting isolated and temporary habitat patches, such as coastal supratidal rockpools, dispersal, whether active or passive, is especially relevant to avoid unfavourable environmental conditions, such as temporary desiccation, and colonise new habitats [1]. In insects with aquatic adults (i.e., water beetles and bugs), flight is a resilient survival strategy that enables them to withstand extended periods of drying by taking into the air and seeking more permanent water, thereby allowing habitat recolonisation and population recovery after water return or refilling [2,8]. Moreover, flight is the primary escape response under extreme environmental stressors [8], triggered by elevated air temperatures [9,10,11] and decreasing water levels [12].

Flight dispersal, a complex trait encompassing rate, distance, distribution, and timing, is challenging to study directly. Although direct measures of flight dispersal can be achieved in the field (e.g., mark-release-recapture methods, see [13,14]), this method presents serious difficulties for small organisms and species travelling long distances. In such cases, indirect measurements based on wing morphology serve as reliable insights into flight performance [15]. Key wing characteristics, including length, width, and area, are closely linked to flight dispersal capacity [16]. Several studies on different insect orders have explored wing loading (body mass to wing area ratio) and wing aspect ratio (wing length to wing width ratio), e.g., [15,16,17,18,19,20]. Low wing loading correlates with superior flight capacity, as larger wings per unit of weight demand less energy for a given flight distance [15,21,22]. Aspect ratio, reflecting wing shape, significantly influences speed and manoeuvrability, with longer and more slender wings associated with higher acceleration capacity [22] and faster flying, as seen in dipterans and odonatans [23]. However, insects with lower aspect ratios (shorter, more rounded wings) exhibit slower flights but greater manoeuvrability, as observed in most lepidopterans [24,25]. Recognising differences in wing morphology between males and females has been considered as indicative of sex-biassed dispersal, a prevalent phenomenon in insects with significant demographic, ecological, and genetic implications, e.g., [18,20,22,26].

Another approach for indirectly estimating dispersal involves genetic methods, such as biparentally inherited markers like microsatellites, that offer great sensitivity to recent genetic flow and provide contemporary estimates of dispersal [27,28]. Analysing pairwise fixation index (F_ST_) alongside geographical distance can also reveal patterns of population connectivity. In this context, populations in proximity should exhibit higher connectivity and gene flow than more distant populations (i.e., a classical isolation-by-distance model, see [29].

Coastal supratidal rockpools are characterised by great physical and chemical variability on a daily and seasonal basis [30,31,32]. During the summer drying phase, aquatic animals are exposed to high levels of ultraviolet light, elevated temperatures, salinity fluctuations, pH variation, oxygen concentration changes, and rapid water depletion [28,32,33]. These extreme and fluctuating conditions restrict the biota to a handful of species [34]. Notably, certain beetle species from the genus *Ochthebius* (Coleoptera, Hydraenidae) exclusively inhabit these challenging habitats (see [35]). Two species, *Ochthebius quadricollis* Mulsant and *Ochthebius lejolisii* Mulsant and Rey, characterised by their wide distributions spanning the Atlantic and Mediterranean coasts, can coexist in the same localities, including the same rockpools along the southern and south-eastern Mediterranean Iberian coastline [28]. Both species are macropterous with flight capabilities and can move locally among pools by either flying or walking when environmental conditions are unfavourable [11,36,37].

Although flight dispersal among pools (also under laboratory conditions) has been observed in *Ochthebius* species [11], the potential for flight dispersal over larger spatial scales (tens to hundreds of kilometres) may be limited by their very small size (approximately 2 mm). Nevertheless, this limitation can be mitigated by utilising physical and biological agents. Coastal *Ochthebius* may passively disperse via the transport of eggs by endo- and epizoochory with seabirds that frequently occur in rockpools, although no cases of phoresy have been documented in these species [38]. However, passive dispersal through marine currents may be the main method by which these insects disperse as marine plankton over medium to large spatial scales [38,39,40]. Therefore, the population genetic structure of the two *Ochthebius* species was predicted from the general circulation pattern of Mediterranean marine currents and associated oceanic fronts [38]. On the other hand, flight assisted by wind is frequent in small, weekly flying insects [41], and wind direction significantly modulates flying aquatic invertebrate biodiversity and metacommunity organisation [42,43,44]. However, their contribution to the genetic structure of supratidal *Ochthebius* species is still unknown. Active flight, whether unassisted or aided by wind, is likely to play an important role in short-distance dispersal. However, information regarding their flight capacities or the occurrence of sex-biassed dispersal is lacking for inhabitants of these environments, where both sexes are exposed equally to extreme conditions.

Here, by using different and complementary approaches (flight experimental tests, wing morphology, and molecular markers), we aim to: (1) assess and compare the flight behaviour and capacity of two congeneric supratidal rockpool species, *O. quadricollis* and *O. lejolisii*; (2) discern sex-diferences in wing morphology related to flight performance; and (3) evaluate the congruence between the different outcomes obtained from the various methods employed. We expect to see wing morphology differences between species and sexes related to their flight capacity, which should be reflected in interspecific differences in the genetic divergence (F_ST_) at a small scale (less than 100 km). Despite the apparent limited contribution of flight to the overall larger-scale genetic structure of these beetles, we aim to explore whether flight capacity can still be considered a predictor of genetic connectivity at a short-distance scale.

## 2. Methods

### 2.1. Flight Behaviour

We used an experimental approach to explore flight avoidance response to warming in the studied *Ochthebius* species, *O. quadricollis* and *O. lejolisii*. Both species show significant morphological differences that make it easy to distinguish them using a stereomicroscope. *O. lejolisii* is mainly characterised by the rugose surface and the serrate edge of the elytra, non-splitted labrum, sorted legs, and the elytra tip without forming an inner angle. In contrast, *O. quadricollis* has smooth elytra without serration, significantly longer legs than *O. lejolisii*, and a markedly bilobed labrum.

Alive adults from two coastal rockpool locations in Murcia (Spain) were collected: Cala Reona (37°37′04″ N 0°42′47″ W) for *O. quadricollis* and Isla Plana (37°34′29″ N 1°12′56″ W) for *O. lejolisii* (Figure 1). These specimens were transported to the laboratory in small, aerated aquaria containing 2–3 cm of seawater and filter paper as a substrate.

In the laboratory, the specimens underwent a 7-day acclimation period at a temperature of 20 °C and a salinity of 90 gL^−1^ under a 12-h light and 12-h dark cycle within a climate chamber (Sanyo MLR-351). For each species, 10 acclimated specimens (without being identified by their sex) were randomly allocated to experimental aquaria. Each of these aquaria contained 100 mL of 90 gL^−1^ salt solution and an artificial stone that was partially submerged to facilitate the emergence and flight of individuals, thus helping them avoid stressful conditions (Figure 2). These aquaria were placed in a temperature-controlled water bath, and the temperature was increasing at a rate of 1 °C/min, starting at 20 °C and ending at 45 °C. The temperature gradient tested represents a range from the species’ habitual temperatures (20–35 °C) to extreme temperatures (>45 °C), which are close to the upper lethal limits recorded for these species [11]. Each species trial was replicated in three independent aquaria.

During the exposure period, individuals who flew or died and the corresponding water temperature were recorded (Appendix A). To assess interspecific differences, we conducted an ANOVA in R software, version 4.2.2 [45], on the total number of individuals who flew and the count within the 5 °C temperature intervals along the ascending temperature gradient. We did not determine the sex of the specimens that flew since they could not be captured.

### 2.2. Wing Morphology

To assess the flight capacity of both species and sexes, we used wing loading (elytron length/wing area) and wing aspect ratio (wing length/wing width) measures. Elytral length served as a proxy for body mass [19], considering the small size of both species (approximately 2 mm) and the impossibility of obtaining the individual weight accurately. Adult specimens collected in the same localities as those used in the flight behaviour experiments were kept dried in the freezer to kill and preserve them. To determine the sex of each specimen, we placed them in a Petri dish with 70% ethanol and then observed them under a Leica S9E stereomicroscope (Leica Microsystems GmbH, Wetzlar, Germany), looking at the shape of the last abdominal ventrite, which is different in males and females. Subsequently, the right elytron and wing of each beetle were extracted and mounted on a glass slide, with the thoracic insertion pointing to the left. We used a 50% dimethyl hydantoin formaldehyde solution (DMHF) for mounting, dipping the wing and elytron into two drops. The wing was then covered with a coverslip, and the elytron was left immersed in the uncovered DMHF drop to avoid potential damage from the coverslip. For each species, 15 male and 15 female specimens were prepared and photographed using a Leica M165C stereomicroscope (Leica Microsystems GmbH, Wetzlar, Germany), equipped with an integrated camera. The microscope settings included a fixed zoom of 1.65 and 10× eyepieces. Images were imported into the ImageJ software, v1.53 [46], to measure the maximum length of each elytron, the maximum length, the maximum width, and the area of each wing (Figure 3). Elytron length, wing area, and both ratios were analysed by an ANOVA in R software, v.4.2.2 [45], with species as the fixed factor and sex as the nested factor. The post-hoc Tukey test was performed for pairwise comparisons.

### 2.3. Microsatellites

To infer species flight dispersal, we used microsatellite markers through direct comparison of F_ST_ values between neighbouring populations located at a distance less than 100 km. Our analysis focused on 15 localities on the coast of the Iberian Peninsula where *O. quadricollis* and *O. lejolisii* coexist (Appendix A). For each species, five randomly selected specimens per locality were sequenced using nine microsatellite markers (SSR) designed separately for each subgenus [47]. To minimise interferences between markers, we conducted two PCR reactions per species, distributing five markers in one and four in the other. Loci detection was performed using GeneMapper v.5 software [48], and a minimum quality criterion of presence in the samples was established, leaving 6 markers for each species (Appendix A). Genetic distances were calculated using the F_ST_ method of Weir and Cockerham [49] for diploid genomes. The analysis was conducted with the ‘hierfstat’ package [50] in R software, v.4.2.2 [45]. We used the permutation Welch paired t-test to analyse interspecific differences in F_ST_ values (9999 Monte Carlo permutations) with the vegan package [51] libraries “MKinfer” and “jmuOutlier.” Confidence intervals (95%) for pairwise F_ST_ estimates were determined through bootstrapping across loci (100 bootstrap samples).

To explore the isolation-distance effect in each species, pairwise F_ST_ values were linearised by utilising the regression method proposed by Rousset [52]. The linearised F_ST_ values were then visualised alongside logarithmically transformed euclidean geographical distance between populations in R [45] using the ggplot2 package [53] and its extension ggpmisc [54], adjusting both variables with a linear model.

## 3. Results

### 3.1. Flight Behaviour

Both species showed flight dispersal behaviour in response to heating (Figure 4), with approximately 60 to 80% of the total individuals tested (*n* = 10) taking flight and the rest died. No significant differences were found between the two species in terms of the total number of individuals that flew during the exposure period (F = 0.046, *p* = 0.831) or across temperature intervals (Table 1, Figure 4).

### 3.2. Wing Morphology

Elytron length showed significant differences between species and sexes (Figure 5A, Table 2). In both species, females were found to be larger than males, with *O. lejolisii* being larger than *O. quadricollis*. Additionally, females of both species have a larger wing area than males, although no significant differences were observed between species for each sex (Figure 5B). The aspect ratio of the wings remained similar across species and sexes (Figure 5C). However, wing loading differed significantly between species, with *O. quadricollis* exhibiting, on average, lower wing loading than *O. lejolisii*. Moreover, in *O. quadricollis*, females have lower wing loading than males, while *O. lejolisii* did not show significant differences between sexes (Figure 5D).

### 3.3. Genetic Divergences

Overall, in both species, small genetic divergences were observed across all locality pairs, as outlined in Table 3. Notably, instances of panmixia were identified, indicated by non-significantly different F_ST_ values from 0, regardless of geographic distance within the studied scale. For example, Cala Panizo and Cala de las Conchas, separated by less than 5 km and Cala Panizo and Punta del Cocedor, separated by over 98 km, exhibited panmixia. *O. quadricollis* showed a slightly greater mean F_ST_ (0.03780) than *O. lejolisii* (0.01642) (Table 3), although that difference was not significant (t = 1.2504, df = 27.901, *p* value = 0.2215). The linear model assessing pairwise F_ST_ and geographical distance did not reveal a significant correlation for either species *(p* value > 0.237), with similar F_ST_ values regardless of the geographical distance compared and indicating no significant association between genetic differentiation and distance (Figure 6). Confidence intervals overlapped for both species.

## 4. Discussion

Our laboratory experiments revealed a pronounced and similar flight response in both studied *Ochthebius* species when exposed to adverse heating, mirroring the environmental conditions often encountered in their natural habitat. While our study did not find a significant effect of temperature on flight, a previous study reported that both *O. quadricollis* and *O. lejolisii* exhibited temperature thresholds for avoidance responses generally lower than 40 °C [11]. Increasing flight activity with rising temperatures under controlled laboratory conditions was observed in other congeneric saline species, such as *Ochthebius glaber* Montes and Soler and *Ochthebius notabilis* Rosenhauer [10]. Although we did not observe significant differences in flight response between the two species under the laboratory conditions tested, Mirón-Gatón et al. [11] reported a higher tendency for walking avoidance in *O. lejolisii* at lower temperature stress compared with *O. quadricollis* (35.62 versus 38.42 °C, respectively). This difference aligns with their distinct thermal tolerance and microhabitat preferences, where *O. lejolisii* favours smaller, shallower pools located further from the coast and prone to drying out, while *O. quadricollis* prefers larger, deeper pools near the sea [28]. Moreover, *O. lejolisii* demonstrates adaptive behaviours, such as walking away from the pool or seeking refuge beneath sediment or rock crevices [11]. Both species displayed sexual dimorphism concerning their dispersal capacity, with females exhibiting larger bodies (measured as elytron length) and wing area. These sex-related variations in body size and wing morphology can impact flight capacity, with larger insects known for greater energy storage and the ability to cover longer distances [55,56]. Furthermore, lower wing loading, associated with more energetically efficient flight [16,21,22,57], was also observed in females, particularly in *O. quadricollis,* these being probably the best dispersers. Interestingly, both species and sexes shared similar wing aspect ratios, indicating higher (longer and narrower wings) ratios compared with other beetle species typically found in inland saline habitats, including those within the genus *Ochthebius* (*O. glaber* and *O. notabilis*, Pallarés et al., unpublished data) and species from the *Enochrus bicolor* group [19], as well as Corixidae species [20].

For small insects similar in size to *Ochthebius* (like fruit flies), the Reynolds number for wing motion is generally low [58], mainly due to the prevalence of laminar flow around the wings. This suggests that the wing morphology of the studied *Ochthebius* species is more suited for wind-assisted gliding than continuous fast-flapping, favouring an energy-efficient active flight [17].

Dispersal costs and benefits are often asymmetric between sexes, leading to sex-biassed dispersal [59]. The timing of dispersal relative to mating is often of crucial importance [60,61,62,63,64]. Pre-dispersal mating favours female-biassed dispersal, while post-dispersal mating favours male-biassed dispersal [61]. In addition, the evolution of female-biassed dispersal seems to be favoured by the fluctuating environment [60]. Therefore, in the case of the studied *Ochthebius* species, the high temporality of the rockpools could favour female-biassed dispersal.

Our behavioural and morphological results imply that rockpool beetle species have a high likelihood of survival in pools exposed to drying through active flight dispersal at local and small spatial scales (a few km). However, due to their small size, flight assisted by wind becomes more probable at larger scales (tens to a hundred km), as observed in experimental release-and-recapture experiments with tiny fruit flies [14]. While the isolation by distance scenario did not yield statistically significant results, the observed low F_ST_ values suggest high population connectivity at the studied spatial scale that may provide partial support to the stepping-stone mechanism [65]. This mechanism could be crucial for the dispersal success and population persistence of *Ochthebius*. The absence of a detectable influence of distance may be attributed to either a recent range expansion or a notable homogenising effect of gene flow related to dispersal [29]. Notably, a recent expansion into the Mediterranean has been previously suggested for *O. lejolisii*, a species initially considered restricted to the Atlantic Sea [39]. At a larger spatial scale (hundreds of km), Villastrigo et al. [38], using other molecular markers (COI and wingless), found a lack of significant relationships between genetic and geographical distance, suggesting that additional factors played a more important role in the genetic structure of *Ochthebius* populations, such as oceanic currents. The lack of concordance between the genetic divergences and the identified morphological variations among species and sexes in relation to their flight capacities may suggest a potential influence of wind currents at smaller spatial scales. This opens the possibility of contrasting passive dispersal mechanisms operating at different spatial scales. Additional research is required to elucidate the role of wind in their dispersal and its broader implications for the genetic structure, as well as the pattern of sex-biassed dispersal by flight in supratidal *Ochthebius* species.

## Figures and Tables

**Figure 1 insects-15-00140-f001:**
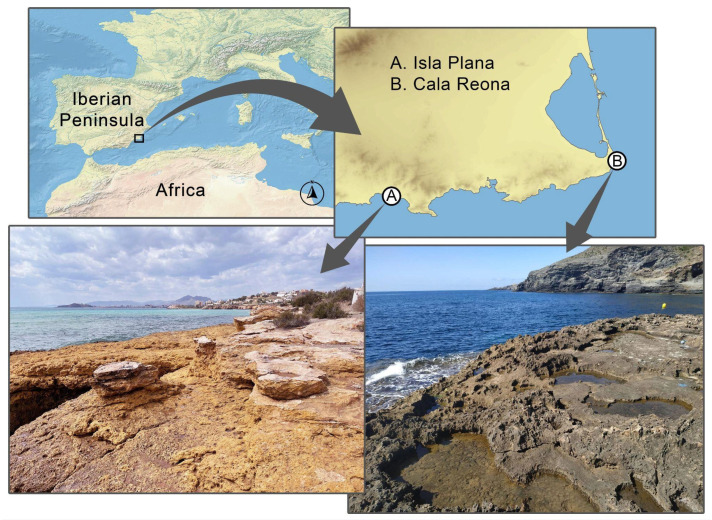
Supratidal rockpool locations used for species collection: (A) Isla Plana; and (B) Cala Reona, Murcia, Spain.

**Figure 2 insects-15-00140-f002:**
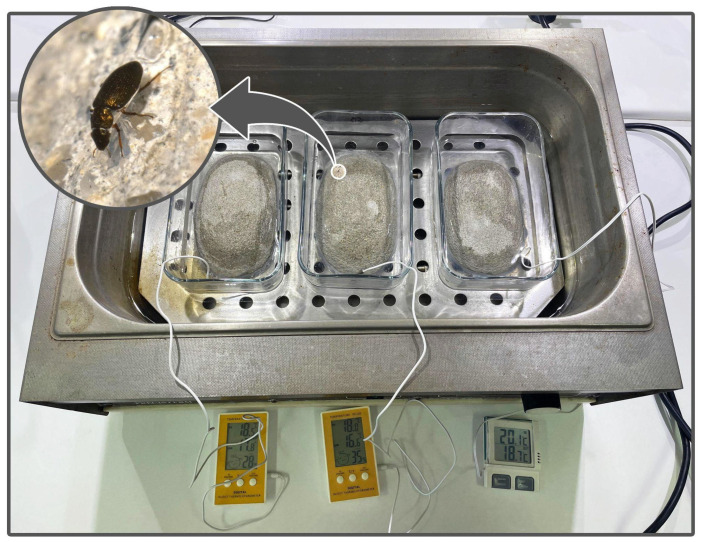
Experimental setup comprising aquaria placed within a water bath under controlled temperatures for flying assays. Each aquaria contained a partially submerged stone to facilitate emergence of individuals from water and dry wings before flying.

**Figure 3 insects-15-00140-f003:**
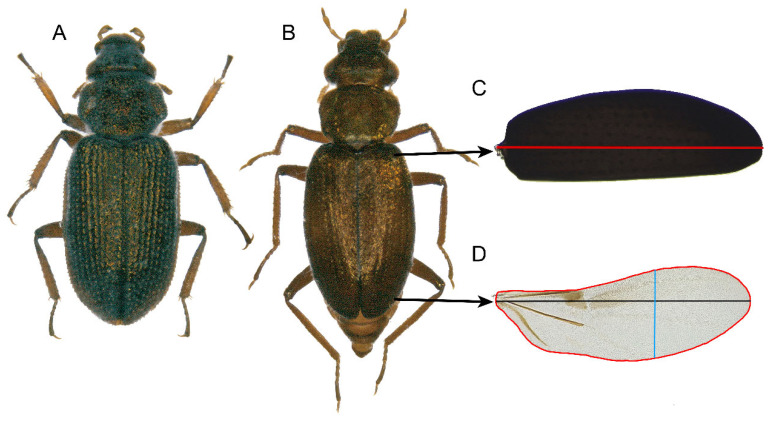
Habitus of *Ochthebius lejolisii* (**A**) and *Ochthebius quadricollis* (**B**), and morphometric measurements made in the right elytron of *O. quadricollis* (**C**, maximum length in red) and in the membranous wing (**D**, perimeter in red, maximum length in black, and maximum width in blue). The same measurements were performed for *O. lejolisii*.

**Figure 4 insects-15-00140-f004:**
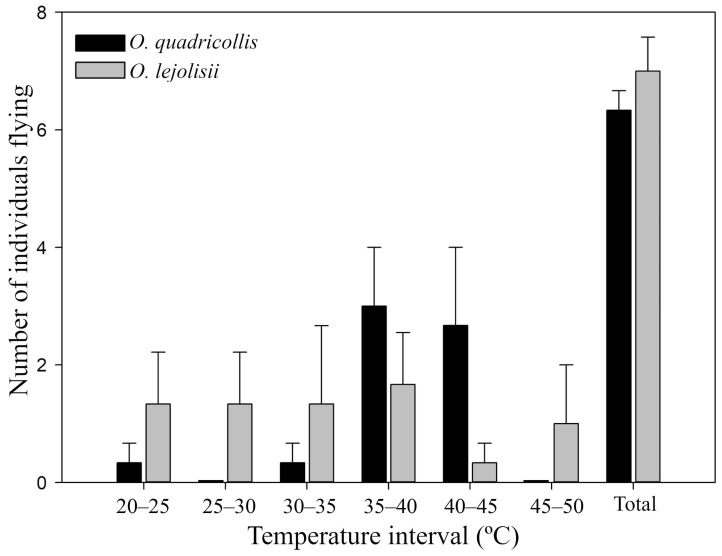
Mean number of individuals flying (+SE) of *Ochthebius quadricollis* and *O. lejolisii* in the experimental trial of increasing temperature (*n* = 10).

**Figure 5 insects-15-00140-f005:**
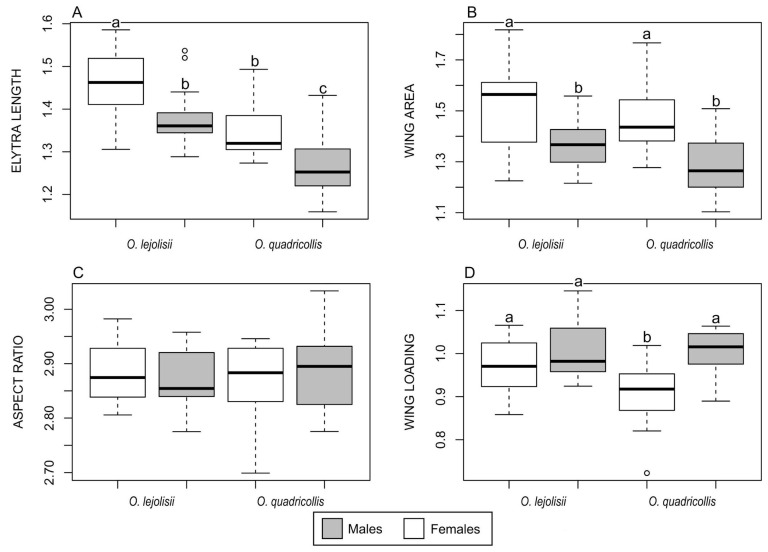
Box plots of the elytron lenght (**A**) and wing metrics (**B**–**D**) in the studied *Ochthebius* species. Box plots depict the minimum, first quartile, median, third quartile, and maximum, with outliers depicted as small circles. Bars with different letters mark significant differences in the pairwise Tukey test (*p* < 0.05).

**Figure 6 insects-15-00140-f006:**
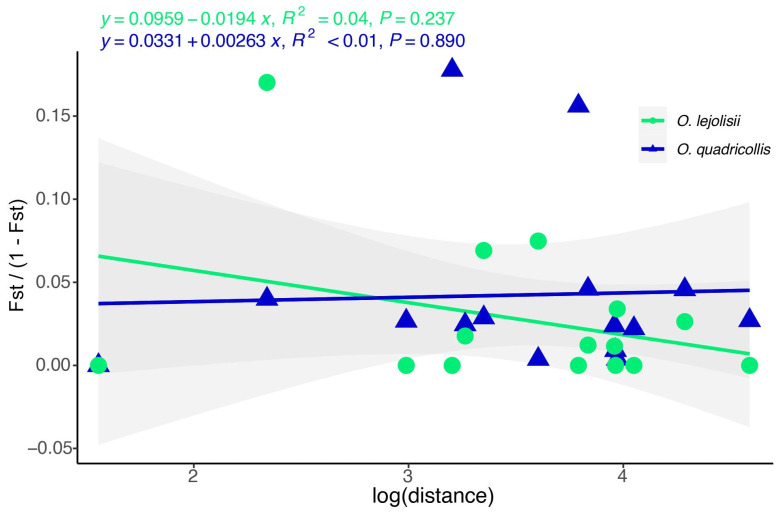
Plot of the relationship between geographical distance and linearised F_ST_ index for the studied species in a small spatial scale (<100 km). Linear tendencies, equations and confidence intervals are shown.

**Table 1 insects-15-00140-t001:** ANOVA results of the effect of temperature and species and their interaction on the flight behaviour of the study *Ochthebius* species.

	Df	Sum	Sq Mean	Sq F value	Pr(>F)
temperature interval	5	14.22	2.844	1.384	0.265
species	1	0.11	0.111	0.054	0.818
temperature × species	5	17.89	3.578	1.741	0.164
Residuals	24	49.33	2.056		

**Table 2 insects-15-00140-t002:** ANOVA results of the effect of species and sex in the elytron and wing metrics on the studied *Ochthebius* species. In bold are significant *p*-values.

	Df	Sum Sq	Mean Sq	F Value	*p* Value
**Elytron length**					
species	1	0.18266	0.18266	34.569	**2.38 × 10^−7^**
sex	2	0.09368	0.04684	8.864	**0.000452**
residuals	56	0.29590	0.00528		
**Wing area**					
species	1	0.0516	0.05155	2.76	0.102
sex	2	0.5178	0.25891	13.86	**1.29 × 10^−5^**
residuals	56	1.0460	0.01868		
**Aspect ratio**					
species	1	0.00087	0.00086	0.186	0.668
sex	2	0.00335	0.00160	0.359	0.700
residuals	56	0.26105	0.00466		
**Wing loading**					
species	1	0.01803	0.01803	4.109	**0.047416**
sex	2	0.07427	0.03713	8.462	**0.000615**
residuals	56	0.24576	0.00439		

**Table 3 insects-15-00140-t003:** Pairwise F_ST_ estimates and 95% confidence intervals obtained from the six microsatellite markers for each species between neighbouring localities (<100 km) on the western Mediterranean coast.

			Pairwise F_ST_ Estimates	
Locality	Locality	Geographic Distance (km)	*Ochthebius quadricollis*	*Ochthebius* *lejolisii*
Moraira	La Illeta	52.40	0.02339 (0.01107–0.03412)	0.01145 (−0.00155–0.03216)
La Illeta	Santa Pola	28.51	0.02796 (−0.00371–0.05866)	0.06467 (0.00333–0.11443)
Santa Pola	Punta del Cocedor	53.07	0.00384 (−0.01610–0.02746)	0.03286 (0.00993–0.05025)
Punta del Cocedor	Cala de las Pulgas	72.69	0.04370 (0.01784–0.07344)	0.02569 (−0.01268–0.08342)
Punta del Cocedor	Cala Panizo	98.31	0.02638 (−0.00920–0.08267)	−0.02076 (−0.03394–−0.01145)
Cala Panizo	Percheles	36.74	0.00377 (−0.03225–0.03935)	0.06962 (−0.05322–0.17334)
Percheles	Cala de las Pulgas	10.39	0.03841 (0.00829–0.07813)	0.14545 (0.04615–0.23690)
Cala de las Pulgas	Cala Panizo	26.14	0.02399 (−0.00029–0.04641)	0.01738 (−0.02107–0.06049)
Cala Panizo	Cala Conchas	4.74	−0.01105 (−0.02868–0.01554)	−0.03440 (−0.11121–0.02991)
Cala Conchas	El Playazo	52.66	0.00901 (−0.01176–0.04150)	0.00000 (0.00000–0.00000)
Cala Panizo	El Playazo	57.36	0.02174 (−0.00796–0.05468)	−0.04519 (−0.07066–−0.02616)
Cala Rijana	Velilla	24.61	0.15088 (0.11918–0.18392)	−0.00709 (−0.03338–0.01618)
Cala Rijana	Nerja	44.30	0.13505 (0.09631–0.17350)	−0.01727 (−0.05207–0.01285)
Velilla	Nerja	19.87	0.02605 (−0.00001–0.05432)	−0.00816 (−0.02670–0.00801)
Cala Milla	Isla de las Palomas	46.33	0.04396 (0.00614–0.08026)	0.01212 (−0.02171–0.03797)
**Mean** **F_ST_** **value**			0.03780	0.01642

## Data Availability

All data produced in this study are available through the main text and the Appendix A.

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
