# Peer review of "Flight Dispersal in Supratidal Rockpool Beetles"

_insects, 2024, doi:10.3390/insects15030140_

Round 1

Reviewer 1 Report

Comments and Suggestions for Authors

I like the combination of data presented in this paper. Adults from two beetle species were collected from rockpools. Behavioural observations provided details of the tendency of the beetles to fly as water temperatures rise. It was unfortunate that the experimental set up did not allow the beetles to be trapped when they flew so they could be sexed. The size of the beetles and their wing area was examined to compare sexes and species. Individuals were genotyped at 6 microsatellite loci from samples from 14 – 16 locations. However, major revision is required as there is a very serious flaw in the theory and analysis of the genetic data which will require substantial reworking.

The authors are not correct in their assertion (line 82) that “one can expect that the sex with higher dispersal will exhibit a lower interpopulation Fst value than the sex that disperses less”. It would be almost impossible to detect a difference in pairwise FST values between males and females of the same two populations because each individual has alleles from both parents – one from a female (mother) and one allele from a male (father). Even if one parent flew and the other never moved – each male and each female offspring would have an equivalent combination of alleles because they both have two parents.  The authors cite a publication I am not familiar with from 1949 but if one consults any modern population genetics text book the error of this approach would be clear.

However, the dataset is interesting and the authors could rescue this paper if they focused on comparing the two species of beetle. Estimates of FST for at least 13 pairwise comparisons are between locations where both species were collected, thus allowing direct comparison of populations at the same geographic location to compare the level of genetic differentiation of the two species in the same landscape.  Given that O. quadricollis females have lower wing loading than either sex of O. lejolisii one could make the prediction that O. quadricollis is capable of greater dispersal which would result in O. quadricollis having lower FST estimates than O. lejolisii when the same locations are compared. The study should combine data for males and females of the same species and calculate pairwise FST (I suggest illustrating this by plotting geographic distance against FST, the two species can be on the same plot with different symbols).  An exact test could be used to test the null (no difference from random in which species has higher or lower FST values), with the prediction that when difference is considered for each pairwise comparison at each location-combination, O. quadricollis will have lower FST estimates than O. lejolisii.

Substantial rewriting of the results and discussion will be required, with likely change to title, abstract and introduction to match. 

Minor

Line 52: active or passive

Line 53: remove “temporary desiccation” as this is a subset of “unfavourable conditions”

Line 58: Do these beetles also fly to avoid predation?

Line 61: delete “fly”

Line 81: delete “nuclear markers” as these are often biparentally inherited markers 

Lines 81-84: delete this comparison (see above)

Lines 128-134: Explain how the two species of beetle can be distinguished morphologically

Line 162: did reference 19 show this to be true or just assume suitability of proxy? Please be explicit in the text

Line 192: Suggest provide justification for sample size of 5 per population

Line 193: explain what you mean by “subgenus”

Line 202: and Figure 4?

Line 248: I don’t understand this section as your results state that flight activity was not observed to be influenced by temperature

Comments on the Quality of English Language

The English is fine but see minor suggestions

Reviewer 2 Report

Comments and Suggestions for Authors

This paper is interesting and well written. These beetles occupy a uniquely stressful environment, and they appear to have evolved a number of adaptations to deal with it. The only difficulty I have with the study is that the beetles are so very small, that some of the studies cited may not be relevant.  I would like to see some assurance that flight characteristics of larger species are applicable to those of this size range and Reynold's number, as this would strenghten the study considerably.

Some references appear to be incomplete, including numbers 23 27, 43, 44  and 46.

Round 2

Reviewer 1 Report

Comments and Suggestions for Authors

Nice to see the change in focus with a new analysis in this manuscript.

I think only minor revision is now needed:

1.     I am uncertain exactly how the mantel tests were performed. Usually with 15 locations, one would construct a pairwise matrix which would contain each sample compared to all other samples (I think that is 99 pairwise comparisons in this instance). The current manuscript has a table with adjacent (neighbouring localities) which has the advantage of reducing pseudo replication but lacks power.  Table 3 data is excellent for paired test of the two species – and for this it is clear that neither species has more genetic structure than the other. In contrast, the mantel test should be repeated for each beetle species using all pairwise comparisons among the 15 population samples.

2.     Line 40 & 314 (and elsewhere): the authors found no evidence of isolation-by-distance and therefore there is no evidence of stepping stone model of gene flow in either beetle species. There are a number of possible explanations which are covered in lines 317 – 319. The study should not conclude that a steeping-stone model can be inferred from their data (for either species) – unless the new mantel tests suggest otherwise.

3.     Line 119: Remove or rewrite this aim: “2) discern patterns of sex-biased flight dispersal;” It is not clear to me that the methods used in this paper would make it possible to discern sex differences. In the behaviour studies of flight the sex of the beetles was not known.

4.     Line 195: Grammar:  Our analysis focused on 15 localities where O. quadricollis and O. lejolisii  coexist on the coast of the Iberian Peninsula (Table S1). 

5.     Is “Fst: the correct format for this journal? Normally “FST [capitals and subscript for “ST”]

6.     Line 255: for any either species 

7.     Legend for table 3 should say “Pairwise FST estimates” and provide the name of the beetle genus and the number of loci used.

8.     I recommend testing to see if each positive pairwise FST estimate differs significantly from zero and adding this information to table 3 (this will help determine whether sample size was large enough).

Author Response

Please Kindly check the attached reply to your comments on the second review. 

Thank you very much for your suggestions.

Jorge Plaza (on behalf of the rest of authors)
